# Relationship of mental health and burnout with empathy among medical students in Thailand: A multicenter cross-sectional study

**Jarurin Pitanupong**[1], **Katti Sathaporn**[1]*, **Pichai Ittasakul**[2],
**Nuntaporn Karawekpanyawong**[3]

**1** Department of Psychiatry, Faculty of Medicine, Prince of Songkla University, Songkhla, Thailand,
**2** Department of Psychiatry, Ramathibodi Hospital, Faculty of Medicine, Mahidol University, Nakhon Pathom,
Thailand, **3** Department of Psychiatry, Faculty of Medicine, Chiang Mai University, Chiang Mai, Thailand

\* katti.s@psu.ac.th

## Abstract

### Objectives

To explore mental health, burnout, and the factors associated with the level of empathy among Thai medical students.

### Background

Empathy is an important component of a satisfactory physician-patient relationship. How-ever, distress, including burnout and lack of personal well-being, are recognized to affect a lower level of empathy.

### Material and methods

A cross-sectional study surveyed sixth-year medical students at three faculties of medicine in Thailand at the end of the 2020 academic year. The questionnaires utilized were: 1) Per-sonal and demographic information questionnaire, 2) Thai Mental Health Indicator-15, 3) The Maslach Burnout Inventory-Thai version, and 4) The Toronto Empathy Questionnaire. All data were analyzed using descriptive statistics, and factors associated with empathy level were analyzed via the Chi-square test or Fisher's exact test, logistic regression., and linear regression.

### Results

There were 336 respondents with a response rate of 70.3%. The majority were female (61.9%). Most participants reported a below-average level of empathy (61%) with a median score (IQR) of 43 (39–40). Assessment of emotion comprehension in others and altruism had the highest median empathy subgroup scores, whereas behaviors engaging higher-order empathic responses had the lowest median empathy subgroup score. One-third of participants (32.1%) had poor mental health, and two-thirds (62.8%) reported a high level of emotional exhaustion even though most of them perceived having a high level of personal accomplishment (97%). The multivariate analysis indicated that mental health was

**Funding:** The authors received no specific funding for this work.

**Competing interests:** All of the authors have declared that no competing interests exist.

statistically significantly associated with the level of empathy. The participants with higher levels of depersonalization had statistically lower scores of demonstrating appropriate sensitivity, altruism, and behaviors engaging higher-order empathic responding.

## Conclusions

Most medical students had below-average empathy levels, and two-thirds of them had high emotional exhaustion levels, yet most of them reported having a high level of personal accomplishment and good mental health. There was an association between mental health and the level of empathy. Higher levels of depersonalization related to lower scores of demonstrating sensitivity, altruism, and behaviors responding. Therefore, medical educators should pay close attention to promoting good mental health among medical students.

## Introduction

Empathy is the capacity to feel or understand what another person is experiencing from within their frame of reference [1]. It is the capacity to place oneself in another's position [2] that facilitates the understanding of the emotions of another person [3, 4]. Nowadays, empathy has been defined as comprising three components: emotional contagion, emotional disconnection, and cognitive empathy [5]. Moreover, empathy is an emotional experience between an observer and a subject in which the observer, based on visual and auditory cues, identifies and transiently experiences the subject's emotional state [6]. To be perceived as empathic, the observer must convey this understanding to the subject. During the initial phase of the process, the observer must not only identify but also understand the basis of the subject's feelings [7].

The importance of empathy should be emphasized throughout medical school because a successful treatment depends in no small measure on effective patient-physician interaction. The physician who understands the patient at a personal level stands a better chance of experiencing and conveying empathy as well as treating the patient effectively than the physician who does not have this level of understanding [8]. Even though empathy is a major component of a satisfactory physician-patient relationship, prior studies suggested that the physician's empathy level might decline throughout clinical training [9]. Four main themes influencing the development and expression of empathy have been identified: subject of medical education curriculum, student's character, patient's profile ("easy" or "difficult" patient), and surrounding conditions [10].

The medical education course taken by the medical student, be it hands-on experience, role models, science and theory, [10] or clinical training can impact empathy negatively [11]. Since the decline in empathy occurs throughout medical school, it can be hypothesized that the training curriculum itself contributes to a decline in compassion among medical students as they go through medical school. Concerning the impact of compassion, student character, insecurities, professional distance [10, 12], mood, maturity, and personal level of empathy are also related factors [10]. Besides, surrounding conditions: time pressure, stress, work environment, and job dissatisfaction are important influences [10]. In Thailand, a prior study found that about one-fifth of medical students had ever thought about dropping out. The most prominent causes of dropout thoughts were studying too hard or disliking the learning environment [13]. Therefore, personal characteristics, learning styles [14], perception of being abused or

mistreated [12, 15, 16], and a hidden curriculum of cynicism may contribute to these problems [17–19]. Distress, which includes burnout [20, 21], depression, anxiety, and lack of personal well-being, has also been recognized to exert an important influence on practice habits [22, 23], performance, or lower levels of empathy among medical students [24, 25].

The Division of Medical Education, Ministry of Education, Thailand proposes the core competencies of medical graduates. According to them, empathy is one constituent of professional habits, communication, and interpersonal skills [26, 27]. However, limited research data concerning this issue exist. To our knowledge, only one study conducted in Thailand in 2012 has explored empathy among medical students. It illustrated that female and preclinical-level students had higher empathy scores than their male and clinical-level counterparts. Notwithstanding, that study did not identify any factors that correlate with empathy level [28]. Therefore, our study aimed to determine whether levels of empathy among medical students are associated with personal distress or burnout and to explore whether a degree of personal well-being or mental health is associated with levels of empathy.

## Materials and methods

### Respondents and procedure

After approval from the corresponding ethics committees of the Faculty of Medicine, Prince of Songkla University (REC: 63-515-3-1), Mahidol University (REC: MURA 2021/52), and Chiang Mai University (REC: 024/2564), this cross-sectional study was conducted on sixth-year medical students attending these three medical schools. The study population comprised all sixth-year medical students of the above-mentioned institutions; they were surveyed at the end of the 2020 academic year. There was a total of 478 sixth-year medical students: 154 from Prince of Songkla University, 174 from Mahidol University, and 150 from Chiang Mai University. To be included, one had to meet the criteria of being a medical student aged more than 20 years and completing all of the questionnaires. Meanwhile, those who were foreign students, declined to participate, or decided to withdraw from the study were excluded. The size of the study was estimated using 57.1% proportion of below-average empathy level from the previous study [25] with 6% margin of error. The sample size required at least 261 for this study.

The data were collected in the given classroom following relevant guidelines by using a paper-based or online process. One participant had the right to choose only one way to answer the questionnaire. Before entering the study, all participants were asked whether responded on paper or online questionnaire to prevent double responses from completing this study. Concerning the paper-based method, a research assistant handed them an information sheet, which delineated the rationale for the study and the allotted time to complete the survey. They had at least 10–15 minutes to consider whether to collaborate in the study or not. If they wished to participate, the research assistant handed them the questionnaires. Adhering to the policy of strict confidentiality, the signatures of the participants were not required, and all the participants retained the right to withdraw from the research at any time. In regards to the online process, all parts of the questionnaire were transformed from their paper form to an online questionnaire using Google Forms. The participants joined the study by either clicking the provided link or scanning a QR code through social media advertisements. Once again, in line with the policy of strict confidentiality, the signatures of the participants were not required, and the participants retained the right to withdraw from the research at any time without giving any reason. Besides, the data were stored in a secure place, and only the researcher could access the information via a password.

## Questionnaire

1. Personal and demographic information questionnaire consisting of questions related to age, gender, religion, hometown, income, cumulative GPA, medical school, history of substance use, physical or psychiatric illness, and specialty preference.

2. Thai Mental Health Indicator-15 (TMHI-15) consisted of 15 questions. The score of each question ranged from 1 to 4; 1 (never); 2 (rarely); 3 (sometimes); and 4 (always), and the total score was between 15 and 60. The interpretation of the total score was as follows: less than 43 (poor mental health), 44–50 (fair mental health), and 51–60 (good mental health). This tool had a Cronbach's alpha coefficient of 0.7 [29].

3. Thai version of the Maslach Burnout Inventory (MBI) questionnaire [30, 31] consisted of 22 items divided into three dimensions: emotional exhaustion (feelings of being emotionally overextended and exhausted by one's work), depersonalization (unsympathetic and impersonal responses toward the recipients of one's care or service), and personal accomplishment (feelings of competence and achievement in one's work with people) [30]. Emotional exhaustion (EE) subscale ranged from 0 (never) to 6 (every day), with Cronbach's alpha coefficient = 0.9. Depersonalization (DP) subscale ranged from 0 (never) to 6 (every day), with Cronbach's alpha coefficient = 0.7. Personal accomplishment (PA) subscale ranged from 0 (every day) to 6 (never), with Cronbach's alpha coefficient = 0.7. For the emotional exhaustion and depersonalization subscales, higher mean scores corresponded to higher degrees of burnout (emotional exhaustion score: 0–16 = low, 17–26 = moderate, >26 = high; depersonalization score: 0–6 = low, 7–12 = moderate, > 12 = high). Lower mean scores of personal accomplishment correspond to higher degrees of burnout (personal accomplishment score: > 38 = low, 32–38 = moderate, 0–31 = high). The Cronbach's alpha coefficient of each domain in the Thai version of MBI is between 0.65–0.92 [30–32].

4. The Toronto Empathy Questionnaire (TEQ) consisted of 16 questions and employed a 5-point rating scale for each question. Item responses were scored according to the following scale for positively worded items; 0 (never); 1 (rarely); 2 (sometimes); 3 (often); and 4 (always). The same scale was applied to reverse score negatively worded items. The scores of all 16 questions were summed, and they ranged from 0 to 64. Higher scores indicated higher levels of self-reported empathy, while total scores below 45 were indicative of below-average empathy levels. The Cronbach's alpha coefficient for this tool was 0.85. Besides, empathy was divided into six subgroups; perception of an emotional state in another that stimulates the same emotion in oneself; assessment of emotion comprehension in others; assessment of emotional states in others by indexing the frequency of behaviors demonstrating appropriate sensitivity; sympathetic physiological arousal; altruism; behaviors engaging higher-order empathic responding, such as pro-social helping behavior [33].

## Statistical analysis

Descriptive statistics such as proportions, means, standard deviation (SD), median and interquartile range (IQR) were calculated. The Chi-square test (or Fisher's exact test), logistic regression, and linear regression analyses were used to identify associations between demographic characteristics, mental health, and burnout with the level of empathy. The analyses were conducted using R version 3.4.1 (R Foundation for Statistical Computing). Statistical significance was defined as a p-value of less than 0.05.

## Results

### Demographic characteristics

The sixth-year medical students who completed the questionnaire were 336 of the 478 total students that were approached; the response rate was 70.3%. Of them, 154 (45.8%) studied at Prince of Songkla University, 93 (27.7%) studied at Chiang Mai University, and 89 (26.5%) studied at Mahidol University. Demographic characteristics were shown in Table 1. Most of them were female (61.9%) and Buddhist (83.9%). Overall, their mean age was 23.5 ±1.5 years, the accumulative GPA was 3.3 ± 0.4, and the income level was 10,000 baht per month. No statistically significant difference in demographic data (gender, religion, GPA) was observed between the students according to the medical school they attended. Additionally, there was no statistically significant difference in demographic data (gender, religion) between participants and non-participants.

### Mental health

According to the Thai Mental Health Indicator-15 (TMHI-15), 158 (47%) participants had fair mental health. Meanwhile, 108 (32.1%) respondent had poor mental health (Table 1).

### Burnout

The Maslach Burnout Inventory (MBI)-Thai version findings indicated that 211 (62.8%) participants had high emotional exhaustion, and another 167 (49.7%) had high depersonalization scores. Only 85 (25.4%) participants had low depersonalization scores. The median MBI score (IQR) for emotional exhaustion was 32 (21–41), and the median score (IQR) for depersonalization was 12 (6–19). However, no one reported a low level of personal accomplishment; almost all of the respondents (97%) perceived a high level of personal accomplishment (Table 1 and Fig 1). The median score (IQR) for personal accomplishment was 13 (9–18).

### Empathy

The Toronto Empathy Questionnaire results revealed that 205 (61%) participants reported a below-average empathy level (Table 1). The total median TEQ score (IQR) was 43 (39–48). According to the six TEQ subgroups, the result showed that the assessment of emotion comprehension in others and altruism had the highest median TEQ subgroup scores (IQR) [3 (2–3) and 3 (2.6–3.3), respectively], whereas behaviors engaging higher-order empathic responding exhibited the lowest median TEQ subgroup score (IQR) [2 (2–3)] (Table 2). No statistically significant difference in the level of empathy between students from different medical schools was detected (p = 0.355).

### The association of demographic characteristics, mental health, burnout, and level of empathy

To identify factors associated with the level of empathy, demographic characteristics, specialty preference, mental health, and burnout were included in the bivariate analysis. Variables with p-values of less than 0.2 from the bivariate analysis were included in the initial model of the multivariate analysis (Table 3). The results showed no relationship between burnout and the level of empathy. There was only mental health remained statistically significantly associated with the level of empathy (Table 4).

However, three dimensions of burnout (EE, DP, PA) were evaluated to determine if they had an impact on any subgroups of empathy. Data were analyzed using linear

**Table 1.** Demographic characteristics, mental health, burnout, and empathy level (N = 336).

| Variables | Number (%) |
|---|---|
| **Gender** | |
| Male | 128 (38.1) |
| Female | 208 (61.9) |
| **Religion** | |
| Buddhism | 282 (83.9) |
| Others (Islam, Christ, others) | 39 (11.6) |
| No answer | 15 (4.5) |
| **Physical illness** | |
| No | 269 (80.1) |
| Yes | 67 (19.9) |
| **Psychiatric illness** | |
| No | 302 (89.9) |
| Yes | 33 (9.8) |
| No answer | 1 (0.3) |
| **Alcohol drinking** | |
| No | 229 (68.2) |
| Yes | 107 (31.8) |
| **Substance use** | |
| No | 328 (97.6) |
| Yes | 8 (2.4) |
| **Specialty preference** | |
| General / not specified | 68 (20.2) |
| Major | 184 (54.8) |
| Minor | 84 (25.0) |
| **Mental health** | |
| Poor | 108 (32.1) |
| Fair | 158 (47.0) |
| Good | 70 (20.8) |
| **Burnout** | |
| Emotional exhaustion | |
| Low | 50 (14.9) |
| Moderate | 74 (22.0) |
| High | 211 (62.8) |
| No answer | 1 (0.3) |
| Depersonalization | |
| Low | 85 (25.4) |
| Moderate | 83 (24.7) |
| High | 167 (49.7) |
| No answer | 1 (0.3) |
| Personal accomplishment | |
| Low | - |
| Moderate | 9 (2.7) |
| High | 326 (97.0) |
| No answer | 1 (0.3) |
| **Empathy level** | |
| Below (<45) | 205 (61.0) |
| Above (>45) | 131 (39.0) |

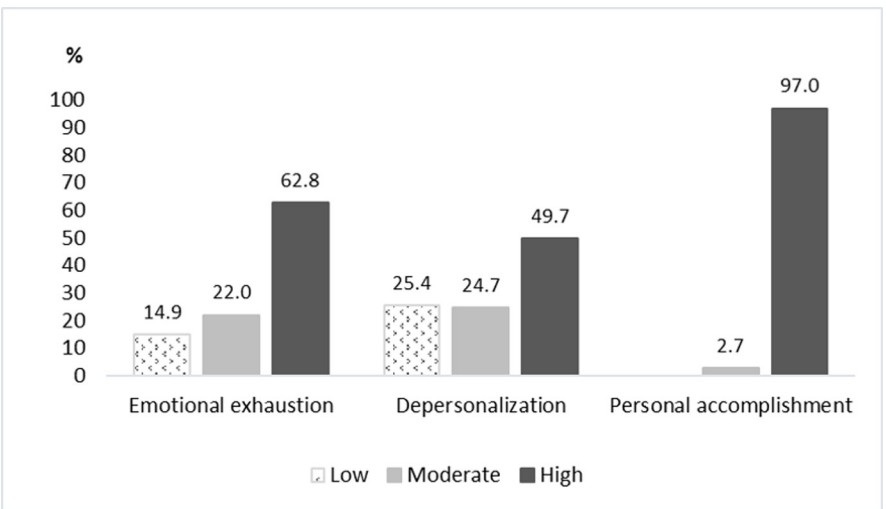

**Fig 1. Burnout rate according to The Maslach Burnout Inventory (MBI)-Thai version.**

regression for each of only two dimensions of burnout (EE and DP) because the majority of participants (97%) reported high PA. It was discovered that participants with higher levels of DP had statistically lower scores of demonstrating appropriate sensitivity, altruism, and behaviors engaging higher-order empathic responding (p<0.001, p<0.003, and p<0.014, respectively).

## Discussion

This study indicated that most sixth-year medical students (61%) understudy had below-average empathy levels, indicated by a median TEQ score (IQR) of 43 (39–48). Regarding empathy subgroups, the assessment of emotion comprehension in others and altruism showed the highest median TEQ subgroup scores, whereas behaviors engaging in higher-order empathic responding exhibited the lowest median TEQ subgroup score. Also, the majority of the respondents reported fair to good mental health; only one-third of them (32.1%) had poor mental health. Moreover, no one perceived a low level of personal accomplishment. Interestingly, most sixth-year medical students perceived high levels of personal accomplishment (97%) even though they reported high levels of emotional exhaustion (62.8%). Besides, the participants with higher levels of depersonalization had statistically lower scores of demonstrating appropriate sensitivity, altruism, and behaviors engaging higher-order empathic responding. There was an association between the level of empathy and mental health. Comparing the level of empathy discovered in our study with those reported by prior research, it was similar to the

**Table 2. Subgroups of empathy according to the Toronto Empathy Questionnaire.**

| Domain of empathy | Median (IQR) |
|---|---|
| Perception of an emotional state in others | 2.5 (2–3) |
| Assessment of emotion comprehension in others | 3 (2–3) |
| Demonstrating appropriate sensitivity | 2.7 (2.4–3) |
| Sympathetic physiological arousal | 2.8 (2.3–3) |
| Altruism | 3 (2.6–3.3) |
| Behaviors engaging higher-order empathic responding | 2 (2–3) |

**Table 3. Bivariate analysis.**

| Variables | Level of empathy Number (%) | | P-value |
|---|---|---|---|
| | <45 | ≥45 | |
| | (N = 205) | (N = 131) | |
| **Gender** | | | 0.213 |
| Male | 84 (41.0) | 44 (33.6) | |
| Female | 121 (59.0) | 87 (66.4) | |
| **Religion** | | | 1 |
| Buddhism | 170 (87.6) | 112 (88.2) | |
| Others (Islam, Christ, others) | 24 (12.4) | 15 (11.8) | |
| **Physical illness** | | | 0.506 |
| No | 167 (81.5) | 102 (77.9) | |
| Yes | 38 (18.5) | 29 (22.1) | |
| **Psychiatric illness** | | | 1 |
| No | 184 (90.2) | 118 (90.1) | |
| Yes | 20 (9.8) | 13 (9.9) | |
| **Alcohol drinking** | | | 0.77 |
| No | 138 (67.3) | 91 (69.5) | |
| Yes | 67 (32.7) | 40 (30.5) | |
| **Substance use** | | | 1[a] |
| No | 200 (97.6) | 128 (97.7) | |
| Yes | 5 (2.4) | 3 (2.3) | |
| **Specialty preference** | | | 0.047 |
| General/not specified | 49 (23.9) | 19 (14.5) | |
| Major | 112 (54.6) | 72 (55.0) | |
| Minor | 44 (21.5) | 40 (30.5) | |
| **Mental health** | | | <0.001 |
| Poor | 90 (43.9) | 18 (13.7) | |
| Fair | 97 (47.3) | 61 (46.6) | |
| Good | 18 (8.8) | 52 (39.7) | |
| **Burnout** | | | |
| Emotional exhaustion | | | 0.007 |
| Low | 25 (12.3) | 25 (19.1) | |
| Moderate | 37 (18.1) | 37 (28.2) | |
| High | 142 (69.6) | 69 (52.7) | |
| Depersonalization | | | <0.001 |
| Low | 40 (19.6) | 45 (34.4) | |
| Moderate | 44 (21.6) | 39 (29.8) | |
| High | 120 (58.8) | 47 (35.9) | |
| Personal accomplishment | | | 0.096[a] |
| Low | - | - | |
| Moderate | 8 (3.9) | 1 (0.8) | |
| High | 196 (96.1) | 130 (99.2) | |

[a] Fisher's exact test

one found by a study from Pakistan [34]. However, our level of empathy was lower than those found in Japanese [35], Turkey [36], and Austrian [37] studies. This variance might be due to differences in study instruments.

**Table 4. The association between mental health, burnout and empathy level (N = 335).**

| Factors | Crude OR (95%CI) | Adjusted OR (95%CI) | P-value LR-test |
| --- | --- | --- | --- |
| **Mental health** | | | <0.001 |
| Poor | Reference | Reference | |
| Fair | 0.32 (0.17,0.58) | 0.35 (0.19,0.65) | |
| Good | 0.07 (0.03,0.14) | 0.07 (0.03,0.16) | |
| **Burnout** | | | |
| Emotional exhaustion | | | 0.522 |
| Low | Reference | Reference | |
| Moderate | 1 (0.49,2.05) | 0.79 (0.34,1.8) | |
| High | 2.06 (1.1,3.84) | 0.62 (0.27,1.44) | |
| Depersonalization | | | 0.088 |
| Low | Reference | Reference | |
| Moderate | 1.23 (0.69,2.33) | 0.99 (0.49,1.98) | |
| High | 2.87 (1.67,4.95) | 1.92 (0.94,3.93) | |
| Personal accomplishment | | | 0.081 |
| High | Reference | Reference | |
| Moderate | 5.31 (0.66,42.93) | 5.54 (0.59,52.45) | |

In terms of empathy subgroups, it is important to keep in mind that empathy encompasses cognitive and affective or emotional dimensions. The cognitive dimension refers to 'the ability to understand the patient's inner experience and perspective, and the capability to communicate this understanding' [38], whereas the affective dimension refers to 'the ability to imagine the patient's emotions and perspectives [39]. In this study, the assessment of emotion comprehension in others and altruism showed the highest empathy subgroup scores, whereas behaviors engaging in higher-order empathic responding exhibited the lowest empathy subgroup score. This might imply that most sixth-year medical students were able to understand the patient's inner experience and imagine their emotions or perspectives. However, they might lack the ability to express empathy toward others. It is, however, agreed that effective interaction or good communication skills on the part of physicians should enable them to convey their actual feelings or experiences to patients. Physicians who are poor communicators and inept at expressing their feelings properly might be misunderstood by patients and the people around them [40]. Moreover, as has been well-established, when patients perceive that the physician understands their conditions and apprehensions, they feel more comfortable with and more willing to confide in the physician. Finally, empathic expression is beneficial to physicians; it reflects that the physician can attune to the psychosocial needs of his/her patient [41].

A prior study identified four main themes that influence the development and expression of empathy: 1) subject or course of study: hands-on experience, role models, science, and theory; 2) students: insecurities and lack of routine, increasing professionalism, previous work experiences, professional distance, mood, maturity, and personal level of empathy; 3) patients: "easy" and "difficult" patients including their state of health; and 4) surrounding conditions: time pressure/stress, work environment, and job dissatisfaction. It suggested that the development and use of empathy could be promoted by increasing hands-on experiences, possibilities to experience the patient's point of view, and offering patient contact early in the curriculum. Besides, medical students need support in reflecting on their actions, behavior, and experiences with patients. Instructors need time and opportunities to reflect on their communication

with and treatment of patients, on their teaching behavior, and their function as role models regard to treating patients empathically and preventing stress [10]. In light of this, it is noteworthy that the current changes implemented in some medical school curriculums in Thailand seem to go in the right direction by integrating patient contact early on in the curriculum and focusing more on teaching adequate communication and interaction behaviors [42]. However, empathy and concern for other's minds may be psychologically distinct and empathy may be limited by our moral lives [43]. Therefore, medical educators should pay close attention to it.

Concerning mental health and burnout, this study determined that lower levels of empathy among medical students were not associated with burnout and that a higher degree of personal well-being or good mental health was also associated with higher levels of empathy. An explanation for this could be the possibility that the majority of our medical students perceived they were in fair to good mental health and perceived a high level of personal accomplishment (97%) even though they reported high levels of emotional exhaustion (62.8%). This could be attributed to the fact that mental health includes having good self-esteem, satisfaction with life, security, confidence in emotional control, being empathetic and happy when helping others, and acknowledging or accepting problems that are difficult to solve [30].

In addition, a prior study on perceived empathy among medical students identified associating factors and sorted them into four clusters: personal experiences, connections, and beliefs; negative feelings and attitudes toward patients; mentoring and clinical experiences that promote professional growth; and school and work experiences that undermine the development of empathy. Hence, it could be said that medical students' experiences that promote personal and professional growth might be the most important factors affecting empathy in medical education. On the other hand, negative feelings and attitudes toward patients, as well as negative school and work experiences, might have a detrimental impact on empathy at all stages of education [42]. Therefore, practical experiences should be made less stressful and strive to promote good mental health among medical students [10].

This study had a few strengths and limitations worth mentioning. To our knowledge, this is the first study that explored mental health and burnout as potential associating factors with the level of empathy among Thai medical students from multiple faculties of medicine. However, this study had some limitations. It was a cross-sectional survey and utilized self-administered questionnaires; therefore, some misunderstandings regarding the intended meaning of questions might have occurred. Nevertheless, to minimize this, questionnaires with good reliability were utilized (good Cronbach's alpha coefficient values). Another drawback was that our data was quantitative, and the sample size was restricted to only medical students who graduated from three medical schools. Additionally, there might have a potential bias due to the greater number of female respondents and a large number of Buddhist respondents. Hence, this dataset may not represent fairly the situation of all Thai medical students in the whole country.

Henceforward, studies are recommended to include all medical students attending all the faculties of medicine in Thailand. In other words, a more comprehensive multi-center study should be conducted. Moreover, other studies should employ more qualitative methods and survey the medical student longitudinally.

## Conclusion

Most medical students had below-average empathy levels, and two-thirds of them had high emotional exhaustion levels, yet most of them reported having a high level of personal accomplishment and good mental health. There was an association between mental health and the level of empathy. Higher levels of depersonalization related to lower scores of demonstrating

sensitivity, altruism, and behaviors engaging empathic responding. Therefore, medical educators should pay close attention to promoting good mental health among medical students.

## Supporting information

**S1 Questionnaire. English version questionnaire.**
(DOC)

**S2 Questionnaire. Thai version questionnaire.**
(DOC)

**S1 Data. English version minimal dataset.**
(XLSX)

## Acknowledgments

The authors appreciative the invaluable contribution of Ms. Kruewan Jongborwanwiwat and Mrs. Nisan Werachattawan related to the data analysis. In addition, we would like to show our gratitude to all the medical students who collaborated in this survey. Moreover, we genuinely appreciate the Faculty of Medicine, Prince of Songkla University, Thailand, and the Office of International Affairs of the Faculty of Medicine for proofreading the English.

## Author Contributions

**Conceptualization:** Jarurin Pitanupong, Katti Sathaporn.

**Data curation:** Jarurin Pitanupong, Katti Sathaporn, Pichai Ittasakul, Nuntaporn Karawekpanyawong.

**Formal analysis:** Jarurin Pitanupong.

**Investigation:** Jarurin Pitanupong, Pichai Ittasakul, Nuntaporn Karawekpanyawong.

**Methodology:** Jarurin Pitanupong.

**Writing – original draft:** Jarurin Pitanupong, Katti Sathaporn, Pichai Ittasakul, Nuntaporn Karawekpanyawong.

**Writing – review & editing:** Jarurin Pitanupong, Katti Sathaporn.

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
