## [Decision Letter · Decision Letter 0]

16 Feb 2022

PONE-D-21-15809Relationship of mental health and burnout with empathy among medical students in Thailand: A multicenter cross-sectional study

PLOS ONE

Dear Dr. Sathaporn,

Thank you for submitting your manuscript to PLOS ONE. After careful consideration, we feel that it has merit but does not fully meet PLOS ONE’s publication criteria as it currently stands. Therefore, we invite you to submit a revised version of the manuscript that addresses the points raised during the review process.

The manuscript has been evaluated by two reviewers, and their comments are available below.

The reviewers have raised a number of major concerns, and have requested additional information on methodological aspects of the study and analyses. Could you please carefully revise the manuscript to address all comments raised?

We look forward to receiving your revised manuscript.

Kind regards,

Vanessa Carels

Staff Editor

PLOS ONE

Journal Requirements:

3. Your abstract cannot contain citations. Please only include citations in the body text of the manuscript, and ensure that they remain in ascending numerical order on first mention.

Reviewers' comments:

Reviewer's Responses to Questions

**Comments to the Author**

1. Is the manuscript technically sound, and do the data support the conclusions?

Reviewer #1: No

Reviewer #2: Yes

2. Has the statistical analysis been performed appropriately and rigorously? 

Reviewer #1: Yes

Reviewer #2: Yes

3. Have the authors made all data underlying the findings in their manuscript fully available?

Reviewer #1: Yes

Reviewer #2: Yes

4. Is the manuscript presented in an intelligible fashion and written in standard English?

Reviewer #1: Yes

Reviewer #2: Yes

5. Review Comments to the Author

Reviewer #1: Dear Authors,

I am delighted to see a study on mental health and burnout from Thailand. Appropriate, well-validated (within a North American context) questionnaires were utilized to assess whether there is an association with empathy, broadly speaking, burnout, mental health, as well as what I perceive to be 'control' variables, such as substance use within your demographics questionnaire. In North American studies, what we do find is generally a decrease in empathy (per scale used for that study), and generally a negative relationship between burnout and empathy. However, given that your study focuses on Thai medical students, these constructs may not hold the same sort of validity, or may need to be interpreted differently. Indications for me was 1) high personal achievement despite burnout, which is fairly surprising given the North American literature, and 2) the conflation of all three aspects of empathy to be equally important, when they may be best served as broken apart into cognitive v emotional empathy. There is a great deal of sociological/philosophical literature and debate on the utilization of emotional empathy (see Paul Bloom's Against Empathy) and whether it is even needed within the clinical encounter.

I recommend to the authors to do their analysis but breaking both burnout and empathy constructs apart into their three dimensions to better understand the reasons for the differences seen. Merely stating that they are cultural differences without pulling in evidence suggestive of such (in Thai, non medical student samples) is far too speculative.

Reviewer #2: Thank you for submitting this manuscript reporting results of your study of three Thai medical school students' self-reported mental health, burnout and empathy. The following comments are meant to help the authors refine their work. I feel these edits will be easy for the authors to make.

Abstract

At the conclusion of your results and in the conclusions, you suggest that the statistical relationship between the mental health scale with empathy is causal. The analysis indicates there is a positive association between the two measures, but it does not mean that better mental health improves empathy levels. The wording needs to be revised to reflect that.

Introduction

Line 96-98: I am not entirely convinced this sentence belongs here. It read as something that should be part of the conclusions of your manuscript and not the background. It may be worth discussing amongst yourselves or the editor to decide.

Materials and Methods

With regard to the administration of the questionnaire, I was curious if there was a plan of who would get the paper questionnaire and who would get the electronic version. The reason I ask is that what if you administer it to a class on paper and the student forgets they did that and completes the electronic version as well. I know it is unlikely, but it is possible. A better explanation of how these two administrations were done and why is needed.

For the section on questionnaires, TMHI-15 and MBI need to have a sentence similar to the TEQ that defines item response options and scores. Along those same lines, it would be helpful to have how many questions on MBI and TEQ are included in their subscores. You indicated scores for MBI, but that does not tell us how many questions nor the scale.

I also had a question about MBI. Is there an expected correlation between the three subscales? It seems that emotional exhaustion and depersonalization would have a positive correlation and the inverse with personal accomplishment. It may be worth noting what should be expected and if your study found similar results.

Results section read nicely.

Discussion

Some additional limitations that should be addressed include potential bias due to greater number of female respondents and large number of Buddhist respondents. Also, have you considered non-response bias for those students not completing the survey?

Conclusion

Line 301 should be modified per my comment about the abstract.

6. PLOS authors have the option to publish the peer review history of their article (what does this mean?). If published, this will include your full peer review and any attached files.

Reviewer #1: No

Reviewer #2: **Yes: **Gary L Beck Dallaghan

---

## [Author Response · Author response to Decision Letter 0]

26 Jul 2022

Dear, editor and reviver team

Thank you for your kindly review, comments, and suggestion.

Author's Responses to Questions of reviewer

Reviewer #1: Dear Authors,

Q: I am delighted to see a study on mental health and burnout from Thailand. Appropriate, well-validated (within a North American context) questionnaires were utilized to assess whether there is an association with empathy, broadly speaking, burnout, mental health, as well as what I perceive to be 'control' variables, such as substance use within your demographics questionnaire. In North American studies, what we do find is generally a decrease in empathy (per scale used for that study), and generally a negative relationship between burnout and empathy. However, given that your study focuses on Thai medical students, these constructs may not hold the same sort of validity, or may need to be interpreted differently. Indications for me was 1) high personal achievement despite burnout, which is fairly surprising given the North American literature, and 2) the conflation of all three aspects of empathy to be equally important, when they may be best served as broken apart into cognitive v emotional empathy. There is a great deal of sociological/philosophical literature and debate on the utilization of emotional empathy (see Paul Bloom's Against Empathy) and whether it is even needed within the clinical encounter.

A: Thanks for your kindly and useful recommendation. 

We remove the sentence or wording that may be too speculative or not relate to the evidence of this research as the following 

This variance might be due to differences in study instruments, population ethnicity, and culture.

We add discussion related to Paul Bloom's Against Empathy [reference 42] as following

However, empathy and concern other’s mind may be psychologically distinct and empathy may be limited by our moral lives [42]. 

Q: I recommend to the authors to do their analysis but breaking both burnout and empathy constructs apart into their three dimensions to better understand the reasons for the differences seen. Merely stating that they are cultural differences without pulling in evidence suggestive of such (in Thai, non-medical student samples) is far too speculative.

A: We try to do more analysis by breaking both empathy constructs apart and burnout into three dimensions as below. 

However, three dimensions of burnout (EE, DP, PA) were evaluated to determine if they had an impact on any subgroups of empathy. Data was analyzed using a linear regression for each of only two dimensions of burnout (EE and DP) because of the majority of participants (97%) reported high PA. It was discovered that participants with higher levels of DP had statistically lower scores of demonstrating appropriate sensitivity, altruism, and behaviors engaging higher-order empathic responding (p<0.001, p<0.003, and p<0.014, respectively).

Reviewer #2: Thank you for submitting this manuscript reporting results of your study of three Thai medical school students' self-reported mental health, burnout and empathy. The following comments are meant to help the authors refine their work. I feel these edits will be easy for the authors to make.

Thanks for your kindness review

Abstract

Q: At the conclusion of your results and in the conclusions, you suggest that the statistical relationship between the mental health scale with empathy is causal. The analysis indicates there is a positive association between the two measures, but it does not mean that better mental health improves empathy levels. The wording needs to be revised to reflect that.

A: We correct the conclusion according your suggestion as 

Most medical students had below-average empathy levels, and two-thirds of them had high emotional exhaustion levels, yet most of them reported having a high level of personal accomplishment and good mental health. Better mental health resulted a protective factor that improves the level of empathy There was an association between mental health and the level of empathy. Higher levels of depersonalization related to lower scores of demonstrating sensitivity, altruism, and behaviors responding.

Introduction

Q: Line 96-98: I am not entirely convinced this sentence belongs here. It read as something that should be part of the conclusions of your manuscript and not the background. It may be worth discussing amongst yourselves or the editor to decide.

A: We remove all these sentence as suggest.

The findings of this study provide useful information for efforts to enhance empathy, promote well-being, and reduce distress among medical students as well as establish educational programs in the medical curriculum geared at boosting medical professionalism.

Materials and Methods

Q: With regard to the administration of the questionnaire, I was curious if there was a plan of who would get the paper questionnaire and who would get the electronic version. The reason I ask is that what if you administer it to a class on paper and the student forgets they did that and completes the electronic version as well. I know it is unlikely, but it is possible. A better explanation of how these two administrations were done and why is needed.

A: We add the more explanation of these issue as 

“The data were collected in the given classroom following relevant guidelines by using a paper-based or online process. One participant had the right to choose only one way to answer the questionnaire. Before entering the study, all participants were asked whether responded on paper or online questionnaire to prevent double response of completing this study.”

Q: For the section on questionnaires, TMHI-15 and MBI need to have a sentence similar to the TEQ that defines item response options and scores. Along those same lines, it would be helpful to have how many questions on MBI and TEQ are included in their subscores. You indicated scores for MBI, but that does not tell us how many questions nor the scale. 

I also had a question about MBI. Is there an expected correlation between the three subscales? It seems that emotional exhaustion and depersonalization would have a positive correlation and the inverse with personal accomplishment. It may be worth noting what should be expected and if your study found similar results.

A: We add the more explanation of the range of each sub-scores in TMHI, MBI Tool as

2) Thai Mental Health Indicator-15 (TMHI-15) consisted of 15 questions. The score of each question ranged from 1 to 4; 1 (never); 2 (rarely); 3 (sometimes); and 4 (always),

3) Thai version of the Maslach Burnout Inventory (MBI) questionnaire [29,30] consisted of 22 items divided into three dimensions: emotional exhaustion (feelings of being emotionally overextended and exhausted by one’s work), depersonalization (unsympathetic and impersonal responses toward the recipients of one’s care or service), and personal accomplishment (feelings of competence and achievement in one’s work with people) [30]. Emotional exhaustion (EE) subscale ranged from 0 (never) to 6 (every day), Cronbach’s alpha coefficient=0.9. Depersonalization (DP) subscale ranged from 0 (never) to 6 (every day), Cronbach’s alpha coefficient=0.7. Personal accomplishment (PA) subscale ranged from 0 (every day) to 6 (never), Cronbach’s alpha coefficient=0.7.

Results 

Q: section read nicely.

A: We feel good, thank you so much ^__^

Discussion

Q: Some additional limitations that should be addressed include potential bias due to greater number of female respondents and large number of Buddhist respondents. Also, have you considered non-response bias for those students not completing the survey?

A: We add the limitation according to suggestion as

Additionally, there might have potential bias due to greater number of female respondents and large number of Buddhist respondents.

Q: Also, have you considered non-response bias for those students not completing the survey?

A: We considered non-response bias for those students who not completing the survey, then we add more explanation in difference of demographic characteristic as 

Additionally, there was no statistically significant difference in demographic data (gender, religion) between participants and non-participants. 

Conclusion

Q: Line 301 should be modified per my comment about the abstract.

A: We correct the conclusion according the suggestion as

Better mental health was found to be a significant protective factor for improving the level of empathy. There was the association between mental health and the level of empathy. Higher levels of depersonalization related to lower scores of demonstrating sensitivity, altruism, and behaviors engaging empathic responding.

Finally 

We have sent our “Revised manuscript” for re-checked and re-proofreading the English from the Office of International Affairs of the Faculty of Medicine. 

Sincerely thanks for every nice, kind comments and suggestion, they are very useful to improve our manuscript. 

Best regards

---

## [Decision Letter · Decision Letter 1]

18 Oct 2022

PONE-D-21-15809R1Relationship of mental health and burnout with empathy among medical students in Thailand: A multicenter cross-sectional studyPLOS ONE

Dear Dr. Sathaporn,

Thank you for submitting your manuscript to PLOS ONE. After careful consideration, we feel that it has merit but does not fully meet PLOS ONE’s publication criteria as it currently stands. Therefore, we invite you to submit a revised version of the manuscript that addresses the points raised during the review process.

ACADEMIC EDITOR:please respond to the comments to make your research suitable for publicationGrammar revision is requested

We look forward to receiving your revised manuscript.

Kind regards,

Omnia Samir El Seifi, Professor

Academic Editor

PLOS ONE

Reviewers' comments:

Reviewer's Responses to Questions

**Comments to the Author**

1. If the authors have adequately addressed your comments raised in a previous round of review and you feel that this manuscript is now acceptable for publication, you may indicate that here to bypass the “Comments to the Author” section, enter your conflict of interest statement in the “Confidential to Editor” section, and submit your "Accept" recommendation.

Reviewer #3: (No Response)

2. Is the manuscript technically sound, and do the data support the conclusions?

Reviewer #3: Partly

3. Has the statistical analysis been performed appropriately and rigorously? 

Reviewer #3: No

4. Have the authors made all data underlying the findings in their manuscript fully available?

Reviewer #3: No

5. Is the manuscript presented in an intelligible fashion and written in standard English?

Reviewer #3: Yes

6. Review Comments to the Author

Reviewer #3: Major revision is needed

One of the objectives is to determine the association between burnout and empathy: So, it is not true to adjust for burnout in logistic regression. Burnout must be included in the model beside mental health.

It is not true to make two models (one for personal and one for depersonalization): you should make one logistic regression model that includes mental health, depersonalization and personal exhaustion. I mean the relation between burnout and empathy should be clearly reported in the logistic regression.

In your models (table 3 and 4), I noticed that you report one p value; it should be two p values: one for each category as compared with the reference group. Do this for the new model.

The sequence of table is not correct: begin with all the descriptive analysis then move to bivariate and then to multivariate: so, table one should include only descriptive statistics. Put the association just before the last table which is the multiple logistic regression.

If there are new findings, discuss them and include them in the conclusion.

Have you checked for multicollinearity in multiple logistic regression?

Your conclusion does not reflect the findings. Please rewrite it.

After conclusion, write your recommendation according to your findings.

7. PLOS authors have the option to publish the peer review history of their article (what does this mean?). If published, this will include your full peer review and any attached files.

Reviewer #3: **Yes: **SAMI ABDO RADMAN AL-DUBAI

---

## [Author Response · Author response to Decision Letter 1]

9 Nov 2022

Response to review Comments 

Title: Relationship of mental health and burnout with empathy among medical students in Thailand: A multicenter cross-sectional study.

Reviewer #3: Major revision is needed

C: One of the objectives is to determine the association between burnout and empathy: So, it is not true to adjust for burnout in logistic regression. Burnout must be included in the model beside mental health.

It is not true to make two models (one for personal and one for depersonalization): you should make one logistic regression model that includes mental health, depersonalization and personal exhaustion. I mean the relation between burnout and empathy should be clearly reported in the logistic regression.

In your models (table 3 and 4), I noticed that you report one p value; it should be two p values: one for each category as compared with the reference group. Do this for the new model.

The sequence of table is not correct: begin with all the descriptive analysis then move to bivariate and then to multivariate: so, table one should include only descriptive statistics. Put the association just before the last table which is the multiple logistic regression.

A: We have modified the manuscript and tables as suggested (as table 1 show only descriptive data, and table 2,3 show bivariate and then multivariate, respectively) 

C: If there are new findings, discuss them and include them in the conclusion.

A: The finding is not different from the original or previous results.

C: Have you checked for multicollinearity in multiple logistic regression?

A: Yes, we check it. Thanks.

C: If your conclusion does not reflect the findings. Please rewrite it.

A: The finding is the same as the prior results, then the conclusion is the same. 

C: After the conclusion, write your recommendation according to your findings.

A: The finding is the same as the prior results. However, we concern about your valuable suggestion, then we add our recommendation according to our findings in conclusion part “Therefore, medical educators should pay close attention to promoting good mental health among medical students.”

Finally, we are grateful for your good wishes and valuable suggestion. This is helpful so much and encourages us to make this manuscript to get better. We wish to have more best wishes from you again. Thanks.

---

## [Editor Report · Decision Letter 2]

12 Dec 2022

Relationship of mental health and burnout with empathy among medical students in Thailand: A multicenter cross-sectional study

PONE-D-21-15809R2

Dear Dr. Sathaporn,

We’re pleased to inform you that your manuscript has been judged scientifically suitable for publication and will be formally accepted for publication once it meets all outstanding technical requirements.

Kind regards,

Omnia Samir El Seifi, Professor

Academic Editor

PLOS ONE
---

## [Editor Report · Acceptance letter]

16 Dec 2022

PONE-D-21-15809R2 

Relationship of mental health and burnout with empathy among medical students in Thailand: A multicenter cross-sectional study 

Dear Dr. Sathaporn:

I'm pleased to inform you that your manuscript has been deemed suitable for publication in PLOS ONE. Congratulations! Your manuscript is now with our production department. 

Kind regards, 

on behalf of

Professor Omnia Samir El Seifi 

Academic Editor

PLOS ONE